# Trends in Incidence and Mortality of Skin Melanoma in Lithuania 1991–2015

**DOI:** 10.3390/ijerph18084165

**Published:** 2021-04-15

**Authors:** Audrius Dulskas, Dovile Cerkauskaite, Ieva Vincerževskiene, Vincas Urbonas

**Affiliations:** 1Department of Abdominal and General Surgery and Oncology, National Cancer Institute, 1 Santariskiu Str., LT-08406 Vilnius, Lithuania; 2Faculty of Medicine, Vilnius University, M. K. Ciurlionio Str. 21, LT-03101 Vilnius, Lithuania; 3Faculty of Medicine, Lithuanian University of Health Sciences, 9 A. Mickeviciaus Str., LT-44307 Kaunas, Lithuania; dovilek8@gmail.com; 4Laboratory of Clinical Oncology, National Cancer Institute, LT-08406 Vilnius, Lithuania; ieva.vincerzevskiene@nvi.lt (I.V.); vincas.urbonas@nvi.lt (V.U.)

**Keywords:** melanoma, epidemiology, incidence, mortality

## Abstract

Background. We aimed to investigate age-specific and sex-specific incidence trends of melanoma in Lithuania from 1991 to 2015. Methods. Analysis was based on data from the population-based Lithuanian Cancer Registry database for 1991–2015, and 6024 cases of skin melanoma were identified. Age-adjusted rates (ASRs) by sex and age group were calculated. Adjustment for ASRs was done using the old European standard population, where a total of three age groups were considered: 0–39, 40–59 and 60+. Additionally, the annual percent change (APC) was calculated, and 95% confidence intervals for APC were calculated. Results. Between 1991 and 2015, the overall melanoma rates increased by an annual percent change (APC) of 3.9% in men (95% CI, 3.6–4.1%) and 2.3% in women (95% CI, 2.1–2.5%). The highest incidences of new cutaneous melanoma cases were observed between old adults (60+) of both sexes, while the lowest incidence rates were observed in the young adult group (up to 39 years old), with the lowest APC (1.6% in males and 0.4% in females). The overall number of melanoma deaths during 1991 and 2015 increased from 64 to 103 deaths per year, and the age-standardized rate (ASR) increased 1.3 times (from 1.8 to 2.4). Conclusions. The incidence and mortality of skin melanoma seem to be increased in all age groups. These trends indicate that skin protection behavior is not sufficient in our population and more efforts need to be taken in order to decrease incidence and mortality rates.

## 1. Introduction

Despite the variety of available prevention strategies, the incidence of melanoma is increasing throughout the world. It is the 19th most common cancer worldwide [1]. According to GLOBOCAN 2020, Australia and New Zealand are the leading countries with the highest age-standardized melanoma incidence rates, with 30.4 new cases in women and 41.6 new cases in men per 100,000 population in 2020 [2].

However, the highest rates of new cases were in Europe and North America with more than 150,000 and 105,000 new cases, respectively, in 2020 [2]. There were more than 57,000 new deaths from melanoma worldwide in 2020 [2].

Increasing incidence trends of cutaneous melanoma are seen in the majority of countries, including the United States, European countries and the United Kingdom [3,4,5]. However, according to the Skin Cancer Institute of Singapore, melanoma incidence rates reached a plateau in Australia seem to be starting to decrease [6]. This might be related to variability that is likely due to the prevalence of risk factors between the regions and specific population groups. Known risk factors for skin melanoma include environmental and lifestyle factors such as exposure to the sunlight and geographical location. Intermittent sun exposure, sunburn history and persistently changed or changing moles are the major risk factors of cutaneous melanoma [7,8]. Family history of melanoma, immunosuppressive states and sun sensitivity (freckles, blue or green eyes, blond or red hair), as well as older age, also plays a major role in malignant melanoma development [9,10].

The purpose of this study was to examine time trends of incidence and mortality rates of skin melanoma for the period 1991–2015 in Lithuania by sex and age.

## 2. Methods

Ethical approval for the analysis of the population-based cancer registry data was not required.

The study is based on the Lithuanian Cancer Registry database covering a population of around 3 million residents according to the 2011 census. The main sources of data are notifications gathered from all hospitals and diagnostic centers in Lithuania. The data from the Lithuanian Cancer Registry database are publicly available. Additionally, death certificate information and population registry information to verify vital status are available.

The study was based on all cases of primary skin melanoma (International Classification of Disease, Tenth Revision (ICD-10) C43) reported to the Registry during 1991–2015. For the analyses, patients were categorized by age at diagnosis according to World Health Organization standard age group definition (0–39, 40–59 and 60+ years) and by sex.

Age-specific and age-standardized incidence rates were calculated. Standardization was performed using the direct method (European standard population). Age-standardized rates were calculated for all ages combined and age groups. Corresponding population data, by age, sex and year, were available from Statistics Lithuania. Age-adjusted rates (ASRs) by sex and age group were calculated. Adjustment for ASRs was done using the old European standard population, where a total of three age groups were considered: 0–39, 40–59 and 60+. Additionally, the annual percent change (APC) was calculated for trends by means of the generalized linear model using the Joinpoint software, version 4.5.0.0 (National Cancer Institute, Bethesda, MD, USA). For each of the identified trends, we also fit a regression line to the natural logarithm of the rates using a calendar year as a regression variable. The 95% confidence intervals for APCs were calculated as well. Annual percent changes were considered statistically significant if *p* < 0.05. The graphical depiction was implemented by using Microsoft Excel software (Microsoft Corporation, Redmond, WA, USA).

## 3. Results

### 3.1. Incidence

According to Lithuanian Cancer Registry 404 278, 6024 new cutaneous melanoma cases were diagnosed in all age groups from 1991 to 2015 in Lithuania.

Between 1991 and 2015, the overall melanoma rates increased by an annual percent change (APC) of 3.9% in men (95% CI, 3.6–4.1%) and 2.3% in women (95% CI, 2.1–2.5%) (Table 1). It was found that melanoma rates increased among young men (0–39 years old) by 1.6% (95% CI, 1.1–2.1%). The incidence rate of cutaneous melanoma was higher among females, with 95 new cases in 1991 and 207 new cases in 2015; among men, only 50 new cases were diagnosed in 1991 and 139 new cases were diagnosed in 2015 (Table 2). Comparing data between 1991 and 2015, the highest increase in the incidence rate of new cutaneous melanoma cases were observed in old adults (60+) of both sexes, while the lowest increases in incidence rate were observed in the young adult group (up to 39 years old), with the lowest APCs (1.6% in males and 0.4% in females) (Figure 1).

Incidence rate among women increased from 4.6 (in 1991) to 8.7 (in 2015) per 100,000, while in men it increased from 3.4 (in 1991) to 8.8 (in 2015) per 100,000 (Figure 2).

### 3.2. Mortality

During the period from 1991 to 2015, 2126 people died as a result of cutaneous melanoma. The overall number of melanoma deaths during 1991 and 2015 increased from 64 to 103 deaths per year, and the age-standardized rate (ASR) increased 1.3 times (from 1.8 to 2.4).

Melanoma deaths were more common in women, with 43 deaths in 1991, and increased over time, with 54 deaths in 2015. Although deaths were more common in women, total melanoma mortality by ASR increased two times in men (from 1.5 to 3.1) while it decreased in women (from 2.0 to 1.9). The ASR increased 1.9 times in men (from 5.5 to 10.4) and decreased in women (from 7.9 to 7.0) from 1991 to 2019 in 60+ age group (Table 2). The mortality trends with deaths per 100,000 flipped in 1998, when mortality became higher in men (Figure 2).

During the period from 1991 to 2015, mortality rates remained almost the same in young adult (0–39) and middle-aged adult (40–59) groups. The highest mortality rates were among older individuals (age group 60+) and increased over time in both sex groups (Figure 2 and Figure 3).

## 4. Discussion

This study has presented for the first time both mortality and incidence rates for skin melanoma and shown the rising trend in Lithuania from 1991 to 2015. The rise of incidence was observed in both sexes and all age groups. Skin melanoma was more common among males than females, and the rate of increase in the incidence was greater in males too.

Prevention and early detection are essential tools to decrease the incidence and mortality rates of melanoma. The primary and most easily accessible prevention of cutaneous melanoma is the reduction of sunlight exposure. This includes midday sun avoidance, covering of the skin and high sun protective factor (SPF) sunscreen usage [9]. This is most important from early childhood to decrease the cumulative sun exposure [11]. However, there are several theories about the impact of sun exposure on melanoma formation [8,12]. It is thought that different types of melanoma are related to different types of sun exposure, i.e., intermittent versus chronic exposure, because different types of sun exposure lead to different spectra of DNA damage. It was found that chronic sun exposure is related only to lentigo malignant melanoma (LMM) formation in older individuals, while intermittent sun exposure is more important in the pathogenesis of superficial spreading melanoma (SSM) and nodular melanoma (NM) in younger individuals [13]. Sunburn history also plays a major role in the pathogenesis of cutaneous melanoma [8,12]. Thus, it is important to decrease not only chronic but also intermittent sun exposure in order to decrease the risk of formation of different types of cutaneous melanoma. Self-examination is also necessary because the majority of melanoma cases can be discovered by the patient [14]. Secondary prevention is accomplished by early diagnosis and treatment of highly curable cutaneous melanoma [9]. People who are at higher risk need to be examined by primary care physicians periodically because this can improve survival rates [9,15]. Thus, the combination of primary and secondary preventions could improve incidence as well as survival rates.

While there are plenty of melanoma prevention strategies, the incidence of new cases is increasing throughout the world, including in Lithuania. The highest incidence rates were found in Australasia (Australia and New Zealand), North America and Europe, including Eastern Europe, Northern Europe and Central Europe [16]. According to the Surveillance, Epidemiology, and End Results (SEER) program of the National Cancer Institute, the highest melanoma of the skin rates are found in White Hispanic and White non-Hispanic populations [17]. Several studies described the same trend. The study describing data from the National Program of Cancer Registries and Surveillance, Epidemiology, and End Results (NPCR-SEER) found that the proportion of all invasive melanoma diagnoses represented by the White non-Hispanic population in different age groups increased from 78.3% to 97.2% from 2001 to 2015 [18].

Melanoma incidence rates vary among different age groups. Several studies discovered that melanoma incidence rates are the lowest in the young adult population (up to 39 years old), as was found in our study (1.1 and 1.7 ASR in 1991 and 2015 respectively) [19]. Watson et al. found that the incidence rate is lower in the 15–29 years old group, with 3.4 cases per 100,000 population, than in the 30–39 years old group, with 12.8 cases per 100,000 [20]. Moreover, they discovered that the incidence rate was lower in males (3.4 cases per 100,000) than in females (7.8 cases per 100,000) in the same age group (15–29 years old) [19]. Purdue et al. found a different trend. They described the increase in melanoma’s incidence rate in young men through the 1970s; however, in the 1980s, they found that this trend changed and incidence rates decreased from 6.6 to 0.4 estimated annual percentage change (EAPC) [20]. There was a similar pattern in the female population; however, from 1992 to 2004, the incidence rate started to increase again [20]. Paulson et al. described the same pattern, but in a larger population than other studies (998,103 cases) [18]. They also found that in the 0–49 years old group, melanoma was more common among females (53.3–68.7%), while from 50 years old it became more common among males (56.7–66.5%) [18]. However, Paulson et al. discovered that melanoma incidence has been decreasing in adolescence (age, 10–19 years) and young adults (age, 20–29 years) from 2005 to 2015 [18]. An epidemiologic study by Lowe et al. described the increased melanoma incidence rates among middle-aged adults (39–60) from 1970 through 2009 [21]. In adolescents and young adults, early melanomas can be histopathologically misdiagnosed due to the presence of intense inflammation and regression of more than 75% of the whole lesion [22].

The highest melanoma incidence rates are found among older adults, and the incidence rate of melanoma increases with the age [23]. Paulson et al. showed this trend in different ethnicities and different age groups of a population of the United States. The overall incidence rate of cutaneous melanoma increased from 200.1 to 229.1 cases per 1 million person-years between 2006 and 2015. The most significant increase was observed among those aged 40 years or older (APC, 1.9%; 95% CI, 1.4–2.4%) [18]. Paulson showed a decrease in incidence rates among younger individuals in both sexes [18]. The same trend is also seen in Lithuania’s population, with 25 cases in the 0–39 years age group and 216 cases in the >60 years age group in 2015.

The differences in incidence rates among different age groups can be explained by melanoma’s pathogenesis, where the degree of cumulative ultraviolet radiation (UVR) exposure, types of oncogenic drivers and mutational load play a major role, as melanoma tumorigenesis is a multistep process involving accumulation of sequential genetic alterations [24,25]. The prevalence of melanoma among older individuals may be associated with the accumulation of lifelong sun exposure in less susceptible individuals, while the early-onset cutaneous melanomas are more associated with intermittent gene–sun interactions in more susceptible individuals [25,26]. In addition, the sunbathing behavior also plays a major role. It is found that men tend to use sunscreen less than women [27]. Thus, they are more susceptible to UVR, as well as being in an increased risk group for melanoma incidence and mortality.

Reed et al. also found that the melanoma incidence rate has been increasing over time. From 1990 to 1999, the melanoma incidence rate was 16.6 per 100,000 person-years, age- and sex-adjusted, but it increased to 30.8 per 100,000 in the 2000–2012 period [28]. Weir et al. reported that melanoma incidence rates had increased more among women (8.85 to 10.31 per 100,000) than among men (5.62 to 5.64 per 100,000) in the United States population from 1999 to 2006 [24]. We found similarities, as well as differences, in our population. We found an increase in the number of cases over time in all age groups (from 145 cases in 1991 to 346 cases in 2015). However, the incidence rates had increased more among males than females in all age groups from 1991 to 2015 (3.9 APC in males and 2.3 APC in females). The increasing incidence rates over time can be explained by several factors. Firstly, the UVR exposure from artificial sources increases as the availability of sunbeds increases. The sunbed fashion started around the 1980s and has been increasing in popularity over time, along with the increase in the incidence of cutaneous melanoma [24]. In 1992, the International Agency for Research of Cancer (IARC) classified UVR and sunbeds as group 2A carcinogenic agents, which are probably carcinogenic to humans [29]. Despite that, the sunbed use prevalence is still quite high [30]. Secondly, increased incidence rates may be associated with improved diagnostic techniques [23]. The fact that females are more frequently diagnosed with melanoma than males may be associated with women attending screening examinations more frequently than men [31].

Despite the increase in melanoma incidence rates throughout the world, the survival rates have been rising too. Watson et al. found that mortality in males decreased by –2.5 APC in the 30–39 years age group and −1.9 APC in the 40–49 years age group and mortality in females decreased by −2.2 APC in the 15–29 years age group and −1.4 APC in the 40–49 years age group from 1992 to 2012 [20]. Forsea found that mortality rates are higher among men and women across Europe, ranging from 0.9 to 2.8 age-standardized rate, world population (ASWR) in females and from 1.3 to 4.2 age-standardized rate, world population (ASWR) in males [3]. We found increasing mortality rates in both sex groups from 1991 to 2015. Like Forsea, we found the highest mortality rates among males (from 1.5 ASR in 1991 to 3.1 ASR in 2015). Higher melanoma mortality rates among men were also found by Mervic et al. [32]. They described that melanomas on lower extremities are more commonly found in females, which have better prognosis and survival outcomes than tumors located on the trunk, which are more commonly found in males [31]. In addition, it was thought that sex hormones might play a role in the incidence and mortality of melanoma. Hormone therapy and pregnancy impact were explored; however, no strong evidence was found to support this hypothesis [31,32]. While in some countries mortality rates are increasing over time, in others we can see the opposite trend. Increased survival rates may be associated with increased screening in several populations as the melanoma may be detected earlier while the tumor is thin [33]. Although there is a lack of evidence that routine population screening is beneficial, several authors described that it may increase survival rates among several populations, especially high-risk ones [33,34]. It was found that thicker lesions at the time of melanoma diagnosis were associated with an increased risk of melanoma mortality [35]. Curiel-Lewandrowski et al. also described that there were 26% fewer melanoma deaths in screened cases than in unscreened cases within a 5-year period [34].

Global death rate from melanoma for both sexes and all age groups increased from 0.072% (0.061–0.093%) in 1991 to 0.11% (0.079–0.12%) in 2015 with an annual percent change of 0.95% [36]. In comparison, the death rate from melanoma in Lithuania for both sexes and all age groups increase from 0.13% (0.11–0.19%) in 1991 to 0.24% (0.16–28%) in 2015 with an annual percent change of 3.07%.

The major strength of the present study is the fact that it included all the population Cancer Registry data of Lithuania. We have been able, for the first time, to demonstrate trends in both mortality and incidence of skin melanoma for the whole population of Lithuania. In addition, incidence trends by sex and age were analyzed.

Our study has some limitations. Firstly, a true increase may underlie the trend in the incidence and mortality being reported. The fact that both incidence and mortality increased supports a non-artefactual effect. Secondly, the staging is not included. Moreover, comorbidities, drug use and other cofactors are not included in the analysis. Lastly, we have not assessed the incidence and mortality trends according to the site of melanoma and cancer-specific vs. other-cause mortality. This information could give a wider view of possible risk factors and relationships to sun exposure.

## 5. Conclusions

Melanoma incidence rates among all age groups and both sexes continue to increase throughout the world, including in Lithuania, despite the variety of available prevention strategies. While mortality rates are decreasing in some countries, populations in several regions, including Lithuania’s population, have higher mortality rates than several decades ago. These trends indicate that skin protection behavior is not sufficient in several populations and more prevention efforts need to be taken in order to decrease incidence and mortality rates. Thus, sunburn and sun protection behavior should be investigated, and educational information needs to be more accessible to society.

## Figures and Tables

**Figure 1 ijerph-18-04165-f001:**
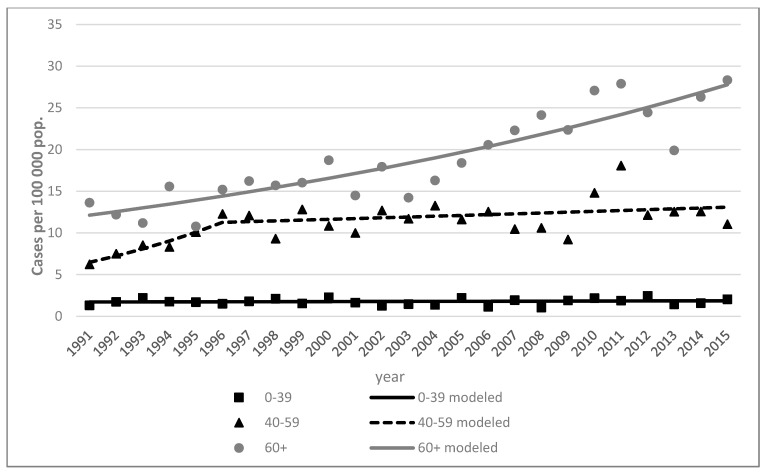
Skin melanoma incidence trends by age in Lithuania, 1991–2015. Women.

**Figure 2 ijerph-18-04165-f002:**
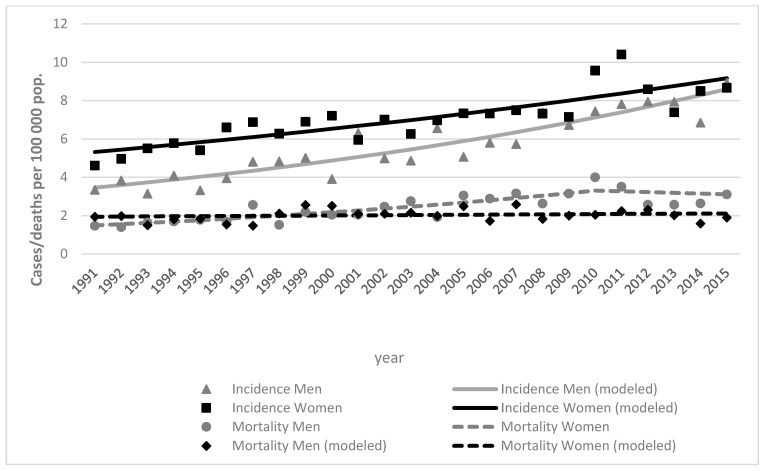
Skin melanoma incidence and mortality trends by sex in Lithuania, 1991–2015, both sexes.

**Figure 3 ijerph-18-04165-f003:**
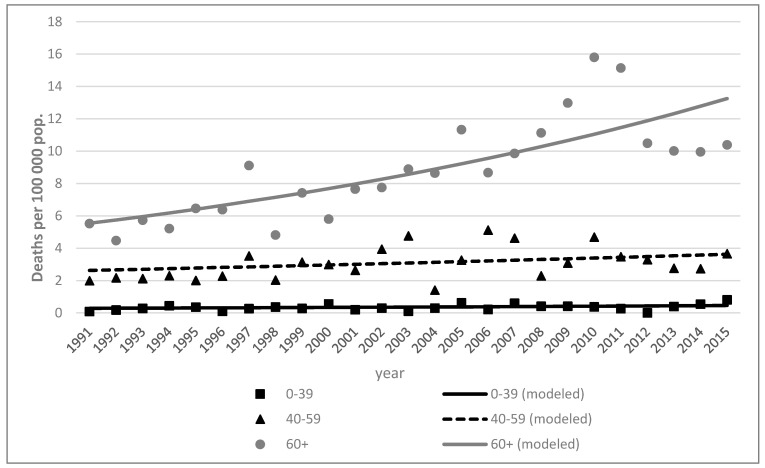
Skin melanoma mortality trends by age in Lithuania, 1991–2015. Men.

**Table 1 ijerph-18-04165-t001:** Results of the Joinpoint regression analysis in skin melanoma incidence trends by sex and age, 1991–2015.

Sex	Age Group	Number of Joinpoints	Line Segment		Annual Percent Change	95% Confidence Intervals	
			Start	End		Lower	Upper
All	All ages	0	1991	2015	2.9 *	2.7	3.1
	0–39	0	1991	2015	0.9 *	0.6	1.2
	40–59	0	1991	2015	2.6 *	2.3	2.9
	60+	0	1991	2015	3.8 *	3.6	4.0
Men	All ages	0	1991	2015	3.9 *	3.6	4.1
	0–39	0	1991	2015	1.6 *	1.1	2.1
	40–59	0	1991	2015	4.0 *	3.6	4.4
	60+	0	1991	2015	4.3 *	4.0	4.6
Women	All ages	0	1991	2015	2.3 *	2.1	2.5
	0–39	0	1991	2015	0.4	−0.1	0.8
	40–59	0	1991	2015	1.8 *	1.4	2.1
		1	1991	1996	11.8 *	7.3	16.4
			1996	2015	0.8 *	0.3	1.2
	60+	0	1991	2015	3.5 *	3.3	3.8

* Annual percent change is statistically significant (*p* < 0.05).

**Table 2 ijerph-18-04165-t002:** Number of skin melanoma cases/deaths and age-standardized mortality rates by sex and age groups in Lithuania, 1991 vs. 2015.

Sex	Age Group	Incidence				Mortality			
		1991		2015		1991		2015	
		Number of Cases	Age-Standardized Rate	Number of Cases	Age-Standardized Rate	Number of Deaths	Age-Standardized Rate	Number of Deaths	Age-Standardized Rate
All	Total	145	4.1	346	8.8	64	1.8	103	2.4
	0–39	24	1.1	25	1.7	2	0.1	6	0.4
	40–59	49	5.6	105	12.2	19	2.2	29	3.3
	60+	72	12.1	216	28.3	43	7.2	68	8.1
Men	Total	50	3.4	139	8.9	21	1.5	49	3.1
	0–39	10	0.9	10	1.4	1	0.1	6	0.8
	40–59	20	5.0	54	13.5	8	2.0	15	3.7
	60+	20	9.4	75	28.4	12	5.5	28	10.4
Women	Total	95	4.6	207	8.7	43	2.0	54	1.9
	0–39	14	1.1	15	1.7	1	0.1	0	0.0
	40–59	29	5.6	51	12.2	11	2.3	14	2.9
	60+	52	12.1	141	28.3	31	7.9	40	7.0

## Data Availability

Data are accessible upon reasonable request to the corresponding author.

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
