# Peer review of "Trends in Incidence and Mortality of Skin Melanoma in Lithuania 1991–2015"

_ijerph, 2021, doi:10.3390/ijerph18084165_

Round 1
Reviewer 1 Report
Thank you for the opportunity to review this article. The manuscript is clearly written and the results are cleary presented. There are several studies with larger cohorts on this topic in the literature. The mortality analysis could increase the originality of the article if a competing risk analysis between specific melanoma deaths vs any other cause was carryed out.
Author Response
Dear Reviewer,
Thank you for your letter and constructive comments concerning our manuscript entitled “Trends in incidence and mortality of skin melanoma in Lithuania 1991-2015”. The paper was revised substantially. Following changes have been made. They are as follows:
Reviewer #1: Thank you for the opportunity to review this article. The manuscript is clearly written and the results are cleary presented. There are several studies with larger cohorts on this topic in the literature. The mortality analysis could increase the originality of the article if a competing risk analysis between specific melanoma deaths vs any other cause was carryed out.
Thank you for great comment. We totally agree that the mortality analysis could increase the originality of the article. However, our death cause analysis is based only on death certificate information. Moreover, sometimes the death cause might be misleading and slightly inaccurate. This was added as our study limitation.
Thank you very much indeed for your comment.
Sincerely
Authors
Reviewer 2 Report
Dear Authors, the present paper is very interesting and a big effort was made to summarize the epidemiologic data.
We just suggest to add some detailed informations in the discussion part on young melanoma and simulators.
lines: Melanoma incidence rates vary among different age groups. Several studies discov-161 ered that melanoma incidence rates are the lowest in young adult’s population (up to 39 162 years old) as in our study (1.1-1.7 ASR in 1991 and 2015 respectively) [20]. .....
"In adolescent and young adults, early melanomas can be histopathologically missdiagnosed due the presence of intense inflammation and regression of more than 75% of the whole lesion"
Rubegni P, Tognetti L, Argenziano G, Nami N, Brancaccio G, Cinotti E, Miracco C, Fimiani M, Cevenini G. A risk scoring system for the differentiation between melanoma with regression and regressing nevi. J Dermatol Sci. 2016 Aug;83(2):138-44.
Author Response
Dear Reviewer,
Thank you for your letter and constructive comments concerning our manuscript entitled “Trends in incidence and mortality of skin melanoma in Lithuania 1991-2015”. The paper was revised substantially. Following changes have been made. They are as follows:
Reviewer #2: Dear Authors, the present paper is very interesting and a big effort was made to summarize the epidemiologic data. We just suggest to add some detailed informations in the discussion part on young melanoma and simulators.
lines: Melanoma incidence rates vary among different age groups. Several studies discov-161 ered that melanoma incidence rates are the lowest in young adult’s population (up to 39 162 years old) as in our study (1.1-1.7 ASR in 1991 and 2015 respectively) [20]. .....
"In adolescent and young adults, early melanomas can be histopathologically missdiagnosed due the presence of intense inflammation and regression of more than 75% of the whole lesion"
Rubegni P, Tognetti L, Argenziano G, Nami N, Brancaccio G, Cinotti E, Miracco C, Fimiani M, Cevenini G. A risk scoring system for the differentiation between melanoma with regression and regressing nevi. J Dermatol Sci. 2016 Aug;83(2):138-44.
Thank you for a great addition – we have included your suggested lines in the Discussion.
Thank you very much indeed for your comment.
Sincerely
Authors
Reviewer 3 Report
Authors reported a really good review of the incidence of melanoma in Lithuania. They con concluded that incidence and mortality of skin melanoma seems to be increased in all age groups. The paper is well written and acceptable for publication
Author Response
A point-by-point response to the editor's comments:
Dear Reviewer,
Thank you for your letter and constructive comments concerning our manuscript entitled “Trends in incidence and mortality of skin melanoma in Lithuania 1991-2015”. The paper was revised substantially. Following changes have been made. They are as follows:
Reviewer #3: Authors reported a really good review of the incidence of melanoma in Lithuania. They con concluded that incidence and mortality of skin melanoma seems to be increased in all age groups. The paper is well written and acceptable for publication
Thank you for your kind comment.
Sincerely
Authors